# Etiological Roles of p75^NTR^ in a Mouse Model of Wet Age-Related Macular Degeneration

**DOI:** 10.3390/cells12020297

**Published:** 2023-01-12

**Authors:** Paula Virginia Subirada, Albana Tovo, María Victoria Vaglienti, José Domingo Luna Pinto, Horacio Uri Saragovi, Maria Cecilia Sánchez, Agustín Anastasía, Pablo Federico Barcelona

**Affiliations:** 1Instituto Ferreyra, INIMEC-CONICET-Universidad Nacional de Córdoba, Córdoba 5016, Argentina; 2Departamento de Bioquímica Clínica, Centro de Investigaciones en Bioquímica Clínica e Inmunología (CIBICI-CONICET), Universidad Nacional de Córdoba, Córdoba 5016, Argentina; 3Centro Privado de Ojos Romagosa, Fundación VER, Córdoba 5000, Argentina; 4Lady Davis Research Institute-Jewish General Hospital, Center for Experimental Therapeutics, Department of Pharmacology and Therapeutics, Department of Ophthalmology and Vision Sciences, McGill University, Montreal, QC H3T 1E2, Canada; 5Instituto Universitario de Ciencias Biomédicas de Córdoba (IUCBC), Córdoba 5016, Argentina

**Keywords:** p75^NTR^, choroidal neovascularization, wet age-related macular degeneration, mononuclear phagocytic cells, neurodegeneration

## Abstract

Choroidal neovascularization (CNV) is a pathological angiogenesis of the choroidal plexus of the retina and is a key feature in the wet form of age-related macular degeneration. Mononuclear phagocytic cells (MPCs) are known to accumulate in the subretinal space, generating a chronic inflammatory state that promotes the growth of the choroidal neovasculature. However, how the MPCs are recruited and activated to promote CNV pathology is not fully understood. Using genetic and pharmacological tools in a mouse model of laser-induced CNV, we demonstrate a role for the p75 neurotrophin receptor (p75^NTR^) in the recruitment of MPCs, in glial activation, and in vascular alterations. After laser injury, expression of p75^NTR^ is increased in activated Muller glial cells near the CNV area in the retina and the retinal pigmented epithelium (RPE)-choroid. In p75^NTR^ knockout mice (p75^NTR^ KO) with CNV, there is significantly reduced recruitment of MPCs, reduced glial activation, reduced CNV area, and the retinal function is preserved, as compared to wild type mice with CNV. Notably, a single intravitreal injection of a pharmacological p75^NTR^ antagonist in wild type mice with CNV phenocopied the results of the p75^NTR^ KO mice. Our results demonstrate that p75^NTR^ is etiological in the development of CNV.

## 1. Introduction

In industrialized countries, age-related macular degeneration (AMD) is one of the most frequent causes of visual impairment and blindness in adults over 60 years [1]. AMD is the deterioration of the macula, the region of the retina that generates the central images of the visual field, and it is associated with risk factors that include genetic background, environmental conditions, and aging [2].

The decrease in retinal function is associated with pathology of the retinal pigmented epithelium (RPE), a monolayer of pigmented cells that forms the external blood–retinal barrier and provides trophic and metabolic support. The RPE is the outermost layer of the retina, and it is sandwiched between photoreceptors that are located in the retina and the choriocapillaris outside the retina. The RPE is separated from the choroid by collagenous and elastic fibers called Bruch’s membrane. The photoreceptors, RPE, Bruch’s membrane, and choriocapillaris are ordered layers, and they constitute a “functional unit” whose organization is crucial for proper vision. This unit is impaired in AMD [3].

Around 15% of patients develop the wet form of AMD, characterized by the growth of new blood vessels from the choroid plexus, a pathological process termed choroidal neovascularization (CNV). The neovascular tufts break through the Bruch membrane and invade the retinal layers, disrupting the RPE and photoreceptor architecture, and inducing neurodegeneration of cones and rods photoreceptors [4]. Therefore, the mechanisms that drive CNV are of relevance to the wet form of AMD, and possibly to other retinal diseases with neovascularization pathologies.

Several reports showed the contribution of mononuclear phagocytic cells (MPCs), such as microglia and infiltrating immune cells, in the progression of CNV in wet AMD patients [5]. At initial stages of choroidal neovascularization, microglia lose their branched morphology and migrate to the afflicted region of the retina and the choroid, and a cytokine cascade recruits additional inflammatory cells, creating a positive feedback loop [5,6], which is interpreted as a chronic state of retinal neuroinflammation [7,8,9].

A mouse model of wet AMD can be induced by laser injury to the outer retina, and the model recapitulates some of the pathology seen in patients [10]. The laser-injured choroid displays an early activation of resident microglia and recruitment of immune cells [11,12]. These cells are relevant to disease progression, as depletion of circulating macrophages or inhibition of macrophage activation reduces the CNV area [13,14,15]. Although inflammation is a key and early component in AMD, the recruitment mechanism of MPCs that results in neovascularization remains cryptic [16,17,18].

We hypothesized that neurotrophin receptors may play a role in wet AMD, as these proteins are documented to play a role in neuroinflammatory ocular diseases, such as optic nerve injury and retinitis pigmentosa, as well as other diseases that also include a vascular pathology component, such as diabetic retinopathy, oxygen-induced-retinopathy, and glaucoma [19]. In these pathologies, after injury, a neurotrophin receptor called p75^NTR^ is mainly expressed in Muller glial cells, where it orchestrates a balance of neuronal survival, neuronal death [20,21,22], or vascular remodelling [23,24].

p75^NTR^ is a transmembrane receptor without intrinsic enzymatic activity, and it is bound by the growth factors neurotrophins or pro-neurotrophins as ligands [25,26]. From the cell membrane, p75^NTR^ transduces intracellular signals that are pleiotropic and, in neurons, can generally lead to atrophy or death. On the contrary, p75^NTR^ can also exert signals in the absence of ligand [27,28,29]. Indeed, as stated, p75^NTR^ plays a key role as the driver of many ocular pathologies, including retinal diseases with inflammatory and neovascularization components [24,30,31,32]. However, the role of p75^NTR^ in wet age-related macular degeneration, and particularly in CNV, is unknown. In a mouse model of wet AMD achieved by laser-induced injury that produces CNV, we sought to determine if p75^NTR^ regulates recruitment of inflammatory cells and angiogenesis, causing CNV and visual impairment.

In the laser CNV model, p75^NTR^ is initially increased in MPCs located in the RPE-Choroid. Biochemical and cytometric analysis suggests that p75^NTR^ participates in the recruitment of infiltrating cells, which are known to promote the angiogenic environment. Genetic ablation (knockout mice) or pharmacological inhibition (antagonism) of p75^NTR^ significantly prevents neovascularization and the consequent functional deficits of the retina, and enables preservation of visual function.

This report contributes to the knowledge of the relevant mechanisms involved in MPCs migration and choroidal neovascularization, and offers a possible strategy to develop therapies.

## 2. Materials and Methods

### 2.1. Mouse Laser-Induced CNV Model

C57BL/6 mice and the p75^NTR^ knockout mice (p75^NTR^ KO) in a C57BL/6 background (kindly provided by Dr. Barbara Hempstead; also available in The Jackson Laboratories: strain #: 002124) were maintained with a 12 h light–dark cycle and with free access to water and food. Mice were handled according to guidelines of the ARVO Statement for the Use of Animals in Ophthalmic and Vision Research. Experimental procedures were designed and approved by the Institutional Animal Care and Use Committee (IACUC) of the Facultad de Ciencias Químicas, Universidad Nacional de Córdoba (Res. HCD 1216/18). All efforts were made to reduce the number of animals used.

The mouse laser-induced CNV model was developed as reported previously [33] and conducted in 2-month-old C57BL/6 and p75^NTR^ KOs; mice with all the manipulation, but without laser, were used as controls. Briefly, mice were anesthetized by inhaled isoflurane, their pupils were dilated with 1% tropicamide, and the cornea was lubricated with gel drops of 0.4% polyethyleneglycol 400 and 0.3% propylene glycol (Systane, Alcon, Buenos Aires, Argentina) to prevent damage. Four laser shots were performed in the posterior retina, using a photocoagulation laser with a slit lamp (wavelength: 532 nm, Iridex, Mountain View, CA, USA) with the following parameters: power—50 mW; duration—100 ms; spot size—100 µm. The visualization of a bubble after the photocoagulation injury was considered a successful spot, as it represents the rupture of the Bruch’s membrane. Burning or hemorrhagic lesions were excluded from the analysis.

### 2.2. Pharmacological Treatment

We carried out pharmacological experiments employing a p75^NTR^ small-molecule antagonist called THX-B (1,3-diisopropyl-1-[2-(1,3-dimethyl-2,6-dioxo-1,2,3,6-tetrahydro-purin-7-yl)-acetyl]-urea) [24,30]. Briefly, mice were topically anesthetized with one drop of proparacaine hydrochloride 0.5% (Anestalcon, Alcon), exophthalmia was induced with one drop of tropicamide 1% (Midril, Alcon, Buenos Aires, Argentina), and eyes were injected at the upper nasal limbus. A single intravitreal injection of THX-B (2 µg total dose) was administered immediately after laser injury. Injection consisted of a 1.0 μL of solution with THX-B (2 µg/µL) diluted in phosphate-buffered saline (PBS), and tissues were collected 4 or 7 days later. Vehicle (PBS)-injected mice were used as controls.

### 2.3. Genotyping

Heterozygous p75^NTR^ knockout mice were intercrossed to yield p75^NTR^+/+, p75^NTR^+/−, and p75^NTR^−/− offspring at Mendelian rates. The p75^NTR^ alleles were identified by PCR analyses of mouse tail genomic DNA obtained at postnatal day 21. The primers employed were the following: P75-1 CGATGCTCCTATGGCTACTA; P75-2 CCTCGCATTCGGCGTCAGCC; P75-3 GGGAACTTCCTGACTAGGGG.

### 2.4. Immunofluorescent Detections in Flat-Mount RPE-Choroids and Retinas

Mice were euthanized at 1, 4, or 7 days after laser and eyes were enucleated and fixed with freshly prepared 4% PFA for 2 h at room temperature (RT). Corneas were detached with scissors along the limbus, and the whole RPE-Choroids and retinas were dissected. Then, they were blocked and permeabilized in Tris-buffered saline (TBS) containing 5% Bovine Serum Albumin (BSA) and 0.1% Triton-X-100 during 6 h at 4 °C. Afterwards, RPE-Choroids were incubated overnight with 0.01 μg/μL of Isolectin IB4 Alexa fluor-488 conjugate (GSA-IB4) from Molecular Probes, Inc. (Eugene, OR, USA), phalloidin-Alexa Fluor 647 (Thermo Fisher Scientific, A22287, Waltham, MA, USA), goat polyclonal anti-p75^NTR^ (1/500; R&D system), mouse anti f4/80 (1/50; Invitrogen, Waltham, MA, USA), rabbit anti CX3CR1 (1/100; Abcam, ab8021, Cambridge, UK), rabbit polyclonal anti-iba1 (1/200; Wako 019-19741), or rabbit polyclonal anti-NG-2 (1/100; AB5320, Millipore, Burlington, MA, USA). RPE-Choroids and retinas were washed with TBS containing 0.1% Triton-X-100 and incubated with secondary antibodies, including against goat, rabbit, or mouse IgG conjugated with Alexa Fluor 488 or 594 (1/250; Molecular Probes, Eugene, OR, USA), during 1 h at RT. The sections were also counterstained with Hoechst 33258 (1:3000; Molecular Probes) for 7 min. After a thorough rinse, the RPE-Choroids and retinas were mounted with Fluor Save (Calbiochem, La Jolla, CA, USA) and cover-slipped. Then, they were flattened by making four radial incisions and mounted with Mowiol 40-88 solution and a glass coverslip to be examined by confocal laser-scanning microscopy (Olympus FluoView FV1200; Olympus Corp., New York, NY, USA). Finally, confocal microphotographs were processed with ImageJ software Version 1.53t (National Institutes of Health, Bethesda, MD, USA). Negative controls without incubation with primary antibody were carried out for each immunodetection (data not shown).

### 2.5. Immunofluorescent Detections in Retinal Cryosections

After mice sacrifice, eyes were enucleated with forceps and then fixed 2 h with 4% PFA during 2 h at room temperature. After that, eyes were incubated overnight in 10%, 20%, and 30% of sucrose in PBS at 4 °C to cryopreserve the tissues. For sections performance, eyes were embedded in optimum cutting temperature (OCT) (Tissue-TEK, Sakura) compound, and 10 μm-thick radial sections were obtained by using a cryostat. The obtained cryosections were stored at −20 °C under dry conditions. Immunostaining was performed as described previously [34]. Briefly, cryosections were washed with PBS, blocked with 2% of BSA in PBS containing 0.1% Tween-20 for 1 h, and then incubated ON at 4 °C with the following primary antibodies, respectively: rabbit polyclonal anti-GFAP (1/1000; Dako, Carpinteria, CA, USA), mouse β3 tubulin (1/500; ab78078, Abcam), and goat anti-p75^NTR^ (1/500; R&D system). Next, sections were washed with PBS 0.1% Tween-20 and incubated with secondary antibodies, including against goat, rabbit, or mouse IgG conjugated with Alexa Fluor 488 or 594 (1/250; Molecular Probes, Eugene, OR, USA), during 1 h at RT. The sections were also counterstained with Hoechst 33258 (1:3000; Molecular Probes) for 7 min. Cryosections were washed twice with PBS 0.1% Tween-20, mounted with Fluor Save (Calbiochem, La Jolla, CA, USA), and cover-slipped. The labeling was captured using a confocal laser-scanning microscope (Olympus Fluvial FV1200; Olympus Corp., New York, NY, USA). Finally, confocal microphotographs were processed with ImageJ software (National Institutes of Health, Bethesda, MD, USA). Negative controls without incubation with primary antibody were carried out for each inmmunodetection (data not shown).

### 2.6. Western Blot

Neural retinas were dissected from RPE/choroid layers, and both tissues were processed separately. Protein extracts were obtained from retinas or RPE-Choroids after homogenization with a lysis buffer that was prepared with 20 mM Tris-HCl pH 7.5, 137 mM NaCl, 2 mM EDTA pH 8, 1% Nonidet P40, 1 mM phenylmethylsulfonyl fluoride (PMSF), 2 mM sodium ortovanadate, and protease inhibitor cocktail (Sigma Aldrich, St. Louis, MO, USA) [35], and later, the tissues were sonicated during 20 s at 40% amplitude. Protein concentration of retinal and RPE-Choroids extracts were quantified d by a DC protein assay kit (Biorad, Hercules, CA, USA), and 50 μg of proteins were electrophoresed in 10% or 15% SDS-PAGE. Then, proteins were transferred to nitrocellulose membranes (Amersham Hybond ECL; GE Healthcare Bio-Sciences AB, Uppsala, Sweden). In order to avoid nonspecific binding of the antibodies, the membranes were blocked with 5% milk in TBS containing 0.1% Tween-20 (TBST) for 1 h at RT. Immediately after, blots were incubated with primary antibodies diluted in 1% BSA in TBST for 1 h overnight at 4 °C. The following antibodies were used: rabbit polyclonal anti-p75^NTR^ (PRB-602C Covance, now Biolegend), rabbit polyclonal anti-GFAP (1/1000; Dako, Carpinteria, CA, USA), mouse monoclonal anti-β tubulin (1/5000; Sigma), mouse monoclonal anti-β-actin (1/2000; ab8226, Abcam). Blots were incubated with IRDye 700 CW donkey anti-rabbit Ig or IRDye 800 CW donkey anti-mouse IgG antibodies (1/15,000 in TBS with 5% BSA) for 1 h, protected from light. After washing with TBST, membranes were visualized using the Odyssey Infrared Imaging System (LI-COR, Inc., Lincoln, NE, USA) and analyzed with ImageJ software (National Institutes of Health, Bethesda, MD, USA).

### 2.7. Electroretinography (ERG)

Electroretinographic signals were measured as previously described [36]. Briefly, 7 days post laser, mice were adapted overnight (ON) in a dark room. The following day, mice were anesthetized intraperitoneally with a solution containing ketamine (90 mg/kg)/xilacine (8 mg/kg), under dim red illumination. The pupils were dilated with 1% tropicamide, and the cornea was lubricated with gel drops of 0.4% polyethyleneglycol 400 and 0.3% propylene glycol (Systane, Alcon, Buenos Aires, Argentina) to protect it from mechanical damage. Then, each mouse was placed in the ganzfeld and exposed to a light stimulus (10 cd·s/m^2^, 0.2 Hz) at a distance of 20 cm. A reference electrode was introduced on the back in the neck, a grounding electrode was placed on the tail, and a gold electrode was located in contact with the central cornea. Electroretinograms were simultaneously recorded from both eyes, and 10 responses to flashes of unattenuated white light from a photic stimulator set at maximum brightness were amplified, filtered (1.5-Hz low-pass filter, 1000 high-pass filter, notch activated), and averaged (Akonic BIO-PC, Buenos Aires, Argentina). The a-wave was determined as the difference in amplitude between the recording at onset and trough of the negative deflection, and the b-wave amplitude was measured from the trough of the a-wave to the peak of the b-wave. The latencies of the a- and b-waves were measured from the time of flash stimulation to the trough of the a-wave or the peak of the b-wave, respectively. Responses were averaged across the two eyes for each mouse.

### 2.8. Fluorescence-Activated Cell Sorting (FACS) Analysis

Retinas and RPE-Choroid from both WT and p75^NTR^ KO mice, with or without laser, were obtained 4 days after the lesion. Under dissecting microscope, retinas and RPE-Choroid were collected in separate tubes and homogenized by gentle pipetting in FACS buffer (cold PBS with 2% FBS, 0.1% sodium azide). Cell suspensions were filtered through a 70 µm cell strainer and washed in FACS buffer. Viability of the cells was checked by Alexa Fluor 700 NHS ester dye (1/15.000; Molecular Probes, Eugene, OR, USA) [37]. Cells were incubated for 30 min at 4 °C with mouse anti f4/80 (1/50; Invitrogen), rabbit anti CX3CR1 (1/100; Abcam, ab8021), FITC rat anti mouse CD11b (1/200; BD Biosciences, Franklin Lakes, NJ, USA), and APC e-Fluor 780 anti-mouse CD45 (1/200; Bioscience). Then, cells were washed with FACS buffer and incubated with goat against mouse IgG conjugated with Alexa Fluor 594 (1/250; Molecular Probes, Eugene, OR, USA) for 30 min at 4°C. FACS of at least 350,000 cells from each RPE-Choroid and 1,000,000 cells from each retina was performed. The labeled cells were analyzed on a BD LSR Fortessa X-20 cytometer (BD Biosciences, Franklin Lakes, NJ, USA) with the FlowJo software (TreeStar), and data are analyzed using FlowJo software (version 7.6.5; FlowJo, Ashland, OR, USA).

### 2.9. Statistical Analysis

Statistical analysis was performed using the GraphPad Prism 5.0 software. A *p*-value < 0.05 was considered statistically significant. Parametric or nonparametric tests were used, according to variance homogeneity evaluated by F or Barlett’s tests. Two-tailed unpaired t or Mann–Whitney tests were used in analysis of two groups. One-way or two-way analysis of variance (ANOVA), followed by Dunnett’s multiple comparison post-test or Kruskal–Wallis, followed by Dunn´s multiple comparison post-test were used to determine statistical significance among more than two different groups. All the assays were performed in *n* ≥ 3 independent experiments, with *n* = 3 technical replicates in each assay, as indicated.

## 3. Results

### 3.1. p75^NTR^ Expression Is Increased in the Choroidal Neovascularization Lesions

First, we quantified the expression of p75^NTR^ in RPE-Choroid tissue in the mouse model of wet AMD. Studies were carried out at 2 time points: 4 days post laser injury (P4 CNV, when the majority of infiltrating cells arrive to the RPE-Choroid) and 7 days post laser injury (P7 CNV, a time point when CNV is fully developed) [10,38].

Western blot analysis showed some expression of p75^NTR^ in healthy RPE-Choroid tissues and a significant increase 4 days after laser injury (Figure 1A). However, the protein levels were non-significantly different between WT and CNV mice 7 days after laser injury (*p* = 0.2286; Figure 1B). The expression of p75^NTR^ in microglia and other mononuclear phagocytic cells was evaluated 4 days after the laser injury. Confocal images of the CNV lesion in choroidal whole mounts showed p75^NTR^ labelling in F4/80 positive cells (Figure 1C) and partially in Iba-1 and CX3CR1 positive microglial cells (Figure 1D,E). For vascular and peri-vascular staining, we performed RPE-Choroid wholemounts 7 days after the laser injury. There was no detectable p75^NTR^ expression in the vasculature, as the p75^NTR^ label was not co-localized with markers of endothelial cells (isolectin IB4) (Figure 1F) and pericytes (NG2) (Figure 1G).

### 3.2. p75^NTR^ Expression Is Also Increased in the Injured Retina

In heathy retinas, p75^NTR^ is expressed at low levels. However, its expression pattern and protein levels may vary after an injury [24]. We evaluated expression of p75^NTR^ in the retinas of CNV mice 7 days after the laser injury, when the photoreceptor layer has been disorganized by neovessels. The RPE-Choroid tissues were excluded, as they were analysed above.

Western blot analysis showed increased expression of p75^NTR^ in retinas CNV mice 7 days after laser, compared to control mice (no laser retinas) (Figure 2A). To further determine which cells residing in the retina increased p75^NTR^ expression, immunostaining of p75^NTR^ was combined with a variety of cellular markers in double fluorescent immunolabelling in cryosections of retina from CNV mice 7 days after laser. There was increased expression of p75^NTR^ in activated Muller glial cells surrounding the lesioned area, identified by glial fibrillary acid protein (GFAP)-positivity and morphology (Figure 2B, upper panel). However, no expression of p75^NTR^ was detected in neurons (Figure 2B, lower panel). Moreover, there was no significant p75^NTR^ staining in pericytes of the superior vascular plexus in retinal whole mounts (Figure 2C).

### 3.3. P75^NTR^ Deletion Reduces the Recruitment of Mononuclear Phagocytic Cells to the Lesioned Area in Laser-Injured Mice

p75^NTR^ has been associated with immune response [39,40,41] and with playing a role in the activation [42] and migration [43,44] of monocytes and immune cells. Therefore, we evaluated if the genetic deletion of p75^NTR^ affected the arrival of MPCs cells to the injured area, using F4/80 flow cytometric analysis.

This quantitative approach allows estimating the percentage of MPCs related to the total CD45 positive cells (myeloid linage) that reach the lesioned area. In the RPE-Choroid suspended cells, we observed an increase in MPCs in WT CNV mice respect to control WT mice 4 days after laser (Figure 3A, upper panels). Notably, p75^NTR^ KO CNV mice exhibited a significant decrease in MPCs population, as compared to WT CNV mice. A similar result was observed in retina tissues (Figure 3A, lower panels). These results suggest that p75^NTR^, which is expressed by MPCs, could have a role in the recruitment of these cells to the lesioned vascular area. There were no statistical differences in the basal percentage of MPCs between WT and p75^NTR^KO mice control (no laser) in the retina or in RPE-Choroid. We did not detect F4/80 immunofluorescence differences in the RPE-Choroids between WT and p75^NTR^KO mice 4 days after laser injury (F4/80 area *p* = 0.1106; F4/80 mean intensity *p* = 0.9833) (Figure 3B). This result suggests that a MPCs subpopulation of F4/80 positive cells is recruited to the injured area.

### 3.4. P75^NTR^ Deletion Reduces Choroidal Neovessels Formation, Lesion Size, and Neuronal Alterations in CNV Mice

As p75^NTR^ expression after laser CNV is incremented in the RPE-Choroid (Figure 1) and in the retina (Figure 2), we investigated a possible role of p75^NTR^ in the formation of choroidal neovessels and the subsequent retinal neurodegeneration in this model. We performed laser lesions in both wild type (WT) and p75^NTR^ knockout (KO) mice to evaluate vascular, glial, and neuronal changes that may occur after laser-induced CNV.

Similar lesions (assessed by the actin-labelled structure of the RPE-Choroid cells) were detected in the RPE-Choroid for WT and p75^NTR^ KO mice 1 day after the laser (Figure 4A), suggesting that genotypes do not interfere with the initial injury by the laser (area of lesion *p* = 0.7519; perimeter of lesion *p* = 0.6713). However, as disease progresses, 7 days after laser, there was a significant reduction in the choroidal neovessel area and perimeter in the p75^NTR^ KO mice, as compared to WT mice (Figure 4B). This result indicates that p75^NTR^ participates in the promotion of CNV development.

Upon damage, macroglial cells in the retina set up a protective response to assure neuronal well-being. The up-regulation of intermediate filaments, hypertrophy, proliferation, and migration of glial cells are key events of reactive gliosis [45]. To evaluate if p75^NTR^ impacts the gliotic response, we measured glial activation by evaluation of GFAP protein levels. Western blot and immunostaining showed an increase in GFAP protein in the WT retina 7 days after laser injury (Figure 4C,D), but no increase in GFAP in the p75^NTR^ KO CNV mice was found, suggesting decreased astrogliosis.

Based on the reduced choroidal neovessel area and the reduced GFAP glial activation of the p75^NTR^ KO after laser injury, we anticipated a healthier neuro-retinal functional unit and improved visual function. This was quantified by scotopic electroretinography studies 7 days after laser injury (Figure 4E). In WT CNV mice, we observed a decreased amplitude of the a- and b-waves, as compared to the control WT control (no laser mice). Remarkably, p75^NTR^ KO CNV mice showed preserved a- and b-waves amplitude, as compared to control p75^NTR^ KO healthy mice. No differences were observed in the a- and b-waves latencies between the experimental groups.

### 3.5. Pharmacological Inhibition of p75^NTR^ Reduced CNV Area and Partially Prevents the Lesion-Induced Retinal Functional Defects

Our previous results suggest that p75^NTR^ has a role in the progression of immune cell recruitment and choroidal neovascularization and in the consequent retinal damage with photoreceptor loss. To further confirm this concept, we carried out pharmacological experiments employing a p75^NTR^ small-molecule antagonist called THX-B (1,3-diisopropyl-1-[2-(1,3-dimethyl-2,6-dioxo-1,2,3,6-tetrahydro-purin-7-yl)-acetyl]-urea) [30]. A single intravitreal injection of THX-B (2 µg total dose) was administered immediately after laser injury, and tissues were collected 4 or 7 days later.

Confocal images of RPE-Choroid flat mounts labelled with isolectin IB-4 showed in the THX-B-treated eyes a significant reduction in CNV area and perimeter 7 days after laser injury (Figure 5A,B).

In functional studies, scotopic electroretinography showed that THX-B injection prevents the decrease in the a-wave amplitude (Figure 5C). However, the injection of THX-B did not prevent the reduction in the b-wave observed in CNV-vehicle mice (Figure 5C). No variations were detected in the a- and b- latency in any of the conditions (Figure 5C).

We further conducted RPE-Choroids cytometric analysis, which showed that in laser-injured mice, THX-B treatment decreased the percentage of MPCs cells, as compared to vehicle treatment. The decreased in MPCs cells in the mice that received THX-B was quantified in the RPE-Choroids (Figure 5D, upper panels) and was quantified also in retinas (Figure 5D, lower panels).

Lastly, THX-B administration did not alter the MPCs fluorescence area and intensity in the RPE-Choroid 4 days after laser injury, as compared with CNV mice with vehicle (F4/80 area *p* = 0.4418; F4/80 mean intensity *p* = 0.1492) (Figure 5E).

Collectively, these p75^NTR^ pharmacological antagonist results are a phenocopy of the results using p75^NTR^ knockout mice.

## 4. Discussion

In this work, we found that p75^NTR^ pharmacological or genetic ablation prevented the neovascularization and the consequent functional deficits of the retina in a murine model of wet age-related macular degeneration (AMD). Moreover, our results suggest that at least part of the p75^NTR^ actions in this model are related to the recruitment of mononuclear phagocytic cells, which are known to promote the angiogenic environment. These results are relevant to understanding the wet AMD pathogenic process and to potentially propose p75^NTR^ as a relevant therapeutic target.

The most widely described effects mediated by p75^NTR^ include neuronal death, long-term depression, and neuronal cytoarchitecture rearrangements [46,47,48,49]. However, the expression of p75^NTR^ and its roles in ocular vasculopathies are less studied. In a recent work, p75^NTR^ KO mice showed a significant reduction in occluded capillaries and a major number of pericytes in the retinas of diabetic mice induced by streptozotocine [50]. In another report, the deletion of p75^NTR^ suppressed the pro-angiogenic factors induced by hypoxia in the RPE [51]. A variety of mechanisms were described to explain these p75^NTR^ effects, ranging from the activation of the transcription factor NF-κB to the cytoplasmic stabilization of HIF-1α by the intracellular fragment of p75^NTR^ generated after proteolytic cleavage [31,52]. In models of retinal vascular disease, increased expression of p75^NTR^ was predominantly in Müller glia, a macroglial cell that is known to secrete pro-angiogenic proteins and to participate in the settlement of the inflammatory response [24], although p75^NTR^ may also be expressed in leucocytes and in infiltrating monocytes after a traumatic injury [53].

Here, in a CNV mouse model, we report expression of p75^NTR^ in MPCs identified as F4/80 positive cells and partially in Iba-1- or CX3CR1-positive cells (Figure 2), suggesting that this receptor is mainly present in MPCs derived from myeloid progenitors. The major expression of the neurotrophin receptor was detected 4 days after the laser injury, coincidental with the arrival of MPCs to the tissue. These protein levels tend to decrease after P4 CNV, with no significant differences between no laser and CNV at P7. Moreover, in the retina, p75^NTR^ expression was restricted to Müller glial cells after the laser-induced lesion. The increased expression of p75^NTR^ in Müller glial cells in the retina and in MPCs in the RPE-Choroid suggest a role of p75^NTR^ in the inflammatory response (Figure 1 and Figure 2). The absence of p75^NTR^ in pericytes and endothelial cells indicates that this receptor is not mediating a direct effect on vascular cells.

We asked whether and how is p75^NTR^ involved in AMD. We focused on MPCs, which we determined are a population expressing the receptor abundantly after the laser. The MPCs population is heterogenous, and they can roughly be classified as tissue-resident or bone-derived, meaning that at least part of them can be recruited from the circulatory system [54]. Classical immunostaining techniques were not sensitive enough to detect differences in MPCs. However, cytometric analysis showed that at least a subpopulation of MPCs were significantly reduced in the injured area of p75^NTR^ KO CNV mice, as compared to WT CNV animals (Figure 3). The difference in the MPCs population resulted in a different inflammatory environment, a key modulator of the angiogenic response. MPC proliferation, activation, and/or recruitment to the lesion area may be directly (as they express p75^NTR^) or indirectly regulated by p75^NTR^. However, we cannot discard that other cells expressing p75^NTR^ may also participate in the settlement and progression of CNV.

The development of neovessels encompasses a strong angiogenic response, mediated by several proteins, including growth factors and lectins, among others. Under different insults, such as hypoxia, hyperglycemia, or chronic inflammation, certain cells residing in both RPE-Choroid and the retina are able to express and secrete pro-angiogenic proteins. However, in many animal models, the blooming of the vascular changes is reached with the arrival of MPCs due to the fact that the inflammation potentiates the response, favoring the secretion of pro-angiogenic molecules [55,56]. In the mouse model of laser-induced CNV, the angiogenic response is known to be regulated by the RPE together with MPCs [17]. Two strategies were employed to determine the participation of p75^NTR^ in CNV: genetic ablation of the receptor utilizing a p75^NTR^ KO mice and pharmacological inhibition by intraocular injection of the p75^NTR^ small-molecule antagonist THX-B.

To reach the RPE-Choroid, the laser beam must go through all the anterior segment, the vitreous, and the retina. We verified that the initial laser lesion areas and perimeters were similar in both mice genotypes, WT and p75^NTR^ KO (estimated with F-actin labelling), suggesting that the initial injury is comparable for both genotypes. Although the initial laser lesion was indistinguishable, vascular outgrowth was reduced in p75^NTR^KO mice 7 days after the laser, indicating that the receptor is involved in the development of CNV. The fact that p75^NTR^ only alters the response of a subgroup of MPCs would explain why there is a partial reduction in the CNV area and perimeter in p75^NTR^KO CNV mice. GFAP expression also provided evidence of reduced retinal damage in p75^NTR^ KO CNV mice, compared to WT CNV mice, 7 days after the laser (Figure 3).

In different retinopathies, such as glaucoma, diabetic retinopathy, optic nerve atrophy, and retinitis pigmentosa, p75^NTR^ is mainly expressed in Muller glial cells [24]. In this regard, it has been reported that the interaction of proNGF with p75^NTR^ in Muller cells increases the production of inflammatory cytokines as tumor necrosis factor α and other proteins as alpha-2 macroglobulin to induce neurodegeneration [21,24,57]. Further experiments should be performed to unravel the role of p75^NTR^ in Muller cells and to unmask the potential ligand involved in its actions in the CNV model.

Electrophysiological studies in CNV animals show that the damage alters retinal functionality, at least in part due to the loss of trophic support provided by the RPE to photoreceptors, as well as the disruption of the retinal architecture by proliferating vessels [58]. Thus, our results suggest that the preserved retinal functionality observed in p75^NTR^ KO CNV mice is a consequence of the reduced neovascular area (Figure 4).

For treatment of neovascular retinopathies, novel therapeutic strategies are needed, especially for patients who become refractory to the classical anti-VEGF therapy. Anti-inflammatory strategies, such as corticosteroid treatment, are one approach [59,60]. Here, the intra-ocular treatment with THX-B demonstrated that p75^NTR^ inhibition reduces the neovessels coverage and the MPCs recruitment, and, importantly, preserved the retinal function (Figure 5). This result suggests that cytokines and other molecules secreted by MPCs regulate the vascular growth and contribute significantly to the development of CNV.

Overall, in this work, we evidenced that p75^NTR^, a receptor frequently linked to neurobiological effects, participates in neovascularization of the choroidal vessels in a clinically relevant model of macular degeneration.

## Figures and Tables

**Figure 1 cells-12-00297-f001:**
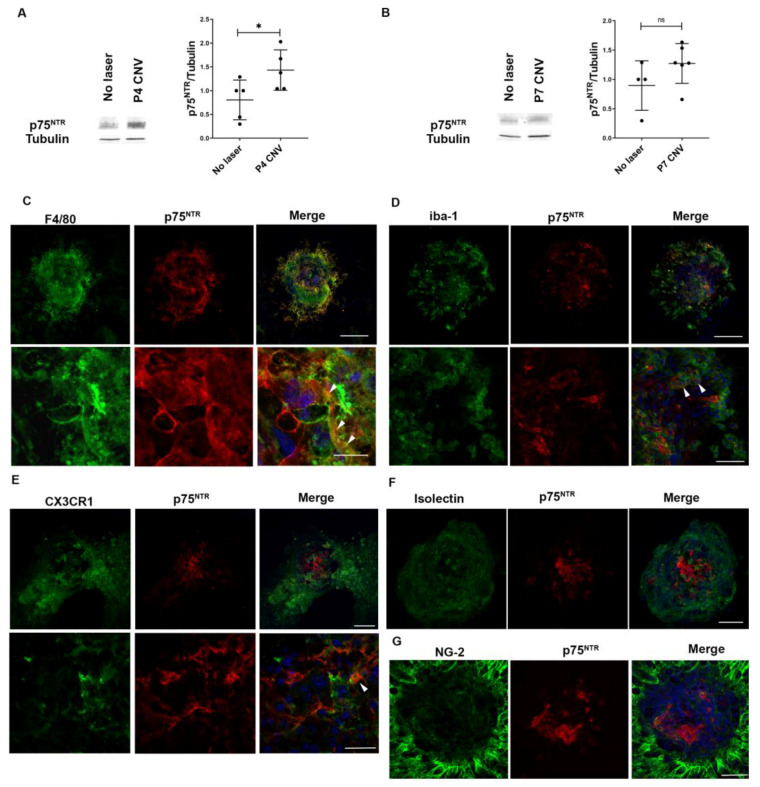
MPCs expressing p75^NTR^ migrate into the laser-injured area of RPE-Choroid. p75^NTR^ co-localizes in F4/80 positive cells in RPE-Choroid. Tissue extracts and flat-mounted retinas were prepared and evaluated by Western blot and by IHC. (**A**) Representative Western blot of RPE-Choroid homogenates prepared from WT mice without CNV, or 4 days after laser injury. Tubulin was used as loading control. Bands were quantified by densitometric analysis, and p75^NTR^/tubulin ratio is represented in the bar graph expressed as units relative to control. Bars denote the mean ± SD from triplicate experiments, *n* = 5 mice/group. The asterisks show statistical differences respect to control. * *p* < 0.05. (**B**) Representative Western blot of RPE-Choroid homogenates prepared from WT mice without CNV, or 7 days after laser injury. Tubulin was used as loading control. Bands were quantified by densitometric analysis, and p75^NTR^/tubulin ratio is represented in the bar graph expressed as units relative to control. ns: non-significant. Bars denote the mean ± SD from triplicate experiments, *n* = at least 4 mice/group. (**C**–**E**) Representative confocal images of RPE-Choroid flat-mounts of WT mice 4 days after laser injury, showing immunofluorescence staining with: (**C**) F4/80 (green), p75^NTR^ (red) and cell nuclei (blue). Scale bar upper panel: 50 µm. Scale bar lower panel: 20 µm. (**D**) IBA-1 (green), p75^NTR^ (red) and cell nuclei (blue). Scale bar upper panel: 100 µm. Scale bar lower panel: 50 µm. (**E**) CX3CR1 (green), p75^NTR^ (red) and cell nuclei (blue). Scale bar upper panel: 100 µm. Scale bar lower panel: 25 µm. (**F**,**G**) Representative confocal images of RPE-Choroid flat-mounts of WT mice 7 days after laser injury, showing immunofluorescence staining with (**F**) Isolectin IB-4 (green), p75^NTR^ (red) and cell nuclei (blue). Scale bar: 50 µm. (**G**) NG-2 (green), p75^NTR^ (red) and cell nuclei (blue). Scale bar: 50 µm.

**Figure 2 cells-12-00297-f002:**
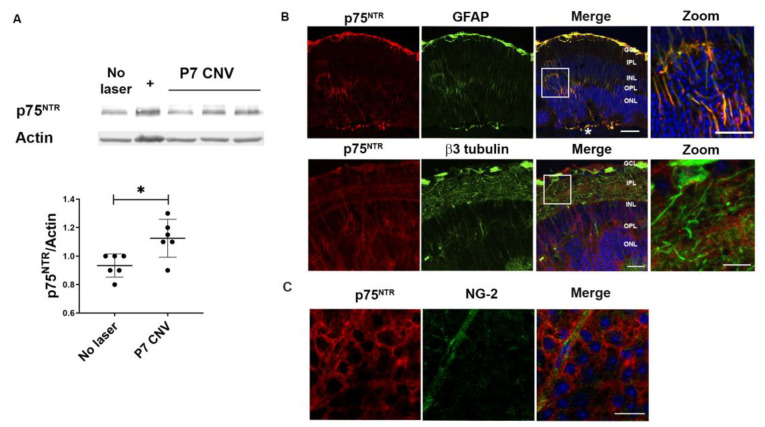
Activated macroglial cells express p75^NTR^ after the laser around the injured area in the retina. p75^NTR^ protein co-localizes with GFAP-positive activated macroglia in CNV mice retinas, 7 days after laser. Tissue extracts and retina sections were prepared and evaluated by Western blot and by IHC. (**A**) Representative Western blot of total retinas homogenates prepared from WT mice without CNV, or 7 days after laser injury. (+) Correspond to a positive control (hippocampus E19 M2). Actin was used as loading control. Bands were quantified by densitometric analysis, and p75^NTR^/actin ratio is represented in the bar graph expressed as units relative to control. Bars denote the mean ± SD from triplicate experiments, *n* = 6 mice/group. The asterisks show statistical differences respect to control. * *p* < 0.05. (**B**) Representative confocal images of WT CNV mice retinal cryosections, 7 days after laser. Upper panel: immunofluorescence staining of GFAP (green) and p75^NTR^ (red). Scale bar: 50 µm. * Indicates the injured area. Zoom scale bar: 25 µm. Cell nuclei counterstained with Hoechst are also shown (blue). Lower panel: immunofluorescence staining of β3 tubulin (green) and p75^NTR^ (red). Scale bar: 25 µm. Zoom scale bar: 10 µm. Abbreviations: GCL (ganglion cells layer), IPL (inner plexiform layer), INL (inner nuclear layer), OPL (outer plexiform layer), ONL (outer nuclear layer). (**C**) Representative confocal images of retinal flat-mount of WT CNV mice, 7 days after laser, showing immunofluorescence staining with NG-2 (green) and p75^NTR^ (red). Scale bar: 50 µm.

**Figure 3 cells-12-00297-f003:**
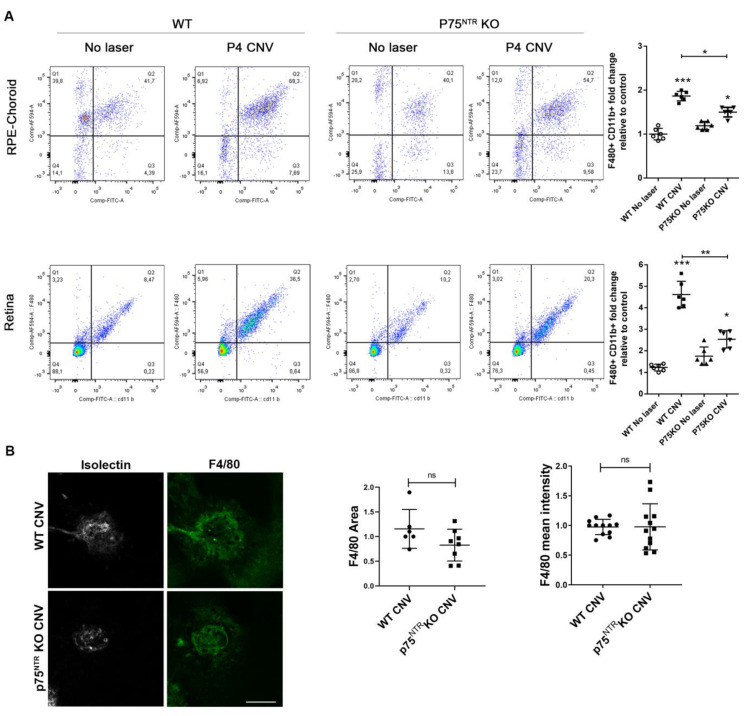
Reduced inflammatory phenotype in the RPE-Choroids and retinas of p75^NTR^ knockout mice, after laser injury. Mononuclear phagocytic cell recruitment is significantly reduced in retinas and RPE-Choroids of p75^NTR^KO mice after CNV. (**A**) Representative flow cytometry pseudocolor plots from WT and p75^NTR^KO mice without CNV, or 4 days after laser injury. Cells in the gate were quantified and the number of cells in the gate/total cells ratio is represented in the bar graph expressed as units relative to WT no laser control. Graphs denote the mean ± SD from triplicate experiments, *n* = 6 mice/group. The asterisks show statistical differences respect to control. * *p* < 0.05, ** *p* < 0.01, *** *p* < 0.001. (**B**) Representative confocal images of RPE-Choroid flat-mounts of WT and p75^NTR^KO mice 4 days after laser, showing immunofluorescence staining with Isolectin IB-4 (grey) and F4/80 (green). Scale bar: 200 µm. F4/80 fluorescence intensity and area were quantified with ImageJ FIJI software Version 1.53t, and represented in the bar graph expressed as units relative to CNV WT control. Bars denote the mean ± SD from triplicate experiments, *n* = at least 5 mice/group. ns: non-significant.

**Figure 4 cells-12-00297-f004:**
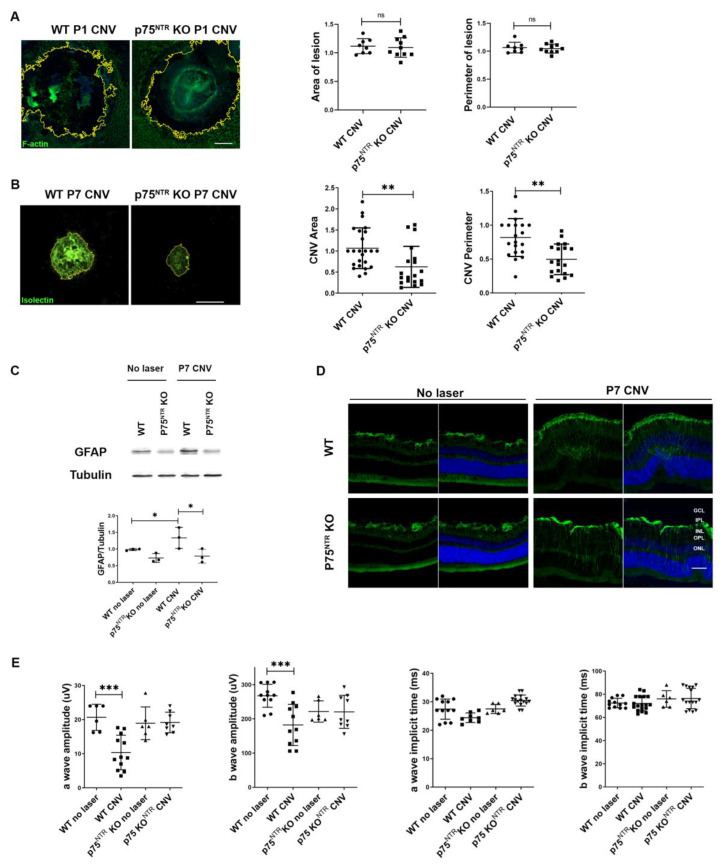
Reduced neovascular phenotype in the choroid of p75^NTR^ knockout mice, after laser injury. The area and the perimeter of choroidal neovessels are significantly reduced in RPE-Choroid flat-mounts of p75^NTR^ KO mice. (**A**) Representative confocal images of RPE-Choroid flat-mounts of WT and p75^NTR^ KO mice 1 day after laser injury, showing immunofluorescence staining with Falloidin (green). The yellow outline represents the lesioned area estimated by F-actin negative staining. Scale bar: 15 µm. Cell nuclei counterstained with Hoechst are also shown (blue). The area and perimeter of the laser lesion were quantified with ImageJ FIJI software, and represented in the bar graph expressed as units relative to WT CNV control. Bars denote the mean ± SD from triplicate experiments, *n* = 4 mice/group. ns: non-significant. (**B**) Representative confocal images of RPE-Choroid flat-mounts of WT mice 7 days after laser injury, showing immunofluorescence staining with Isolectin IB-4 (green). The yellow outline represents the neovessels covered area. Scale bar: 200 µm. The area and the perimeter of the neovessels were quantified with ImageJ FIJI software, and represented in the bar graph expressed as units relative to CNV WT control. Bars denote the mean ± SD from triplicate experiments, *n* = at least 6 mice/group. The asterisks show statistical differences respect to control. * *p* < 0.05, ** *p* < 0.01, *** *p* < 0.001. (**C**) Representative Western blot of total retinal homogenates prepared from WT and p75^NTR^KO mice without CNV, or 7 days after laser injury. Tubulin was used as loading control. Bands were quantified by densitometric analysis, and GFAP/tubulin ratio is represented in the bar graph expressed as units relative to control. Bars denote the mean ± SD from triplicate experiments, *n* = 3 mice/group. (**D**) Representative immunofluorescence analysis of GFAP (green) in retinal cryosections from WT and p75^NTR^KO mice without injury or 7 days after laser injury. Scale bar: 50 µm. Cell nuclei counterstained with Hoechst are also shown (blue). Abbreviations: GCL (ganglion cells layer), IPL (inner plexiform layer), INL (inner nuclear layer), OPL (outer plexiform layer), ONL (outer nuclear layer). (**E**) Amplitudes and implicit times of a- and b-waves from scotopic electroretinograms recorded in WT and p75^NTR^KO mice without injury or 7 days after laser injury. The data show averages of responses of both eyes. Graphs denote the mean ± SD from triplicate experiments, *n* = at least 6 mice/group. The symbol correspond: WT no laser (filled circle), WT CNV (filled square), p75^NTR^KO no laser (filled triangle up), p75^NTR^KO CNV (filled triangle down). The asterisks show statistical differences respect to control. * *p* < 0.05, ** *p* < 0.01, *** *p* < 0.001.

**Figure 5 cells-12-00297-f005:**
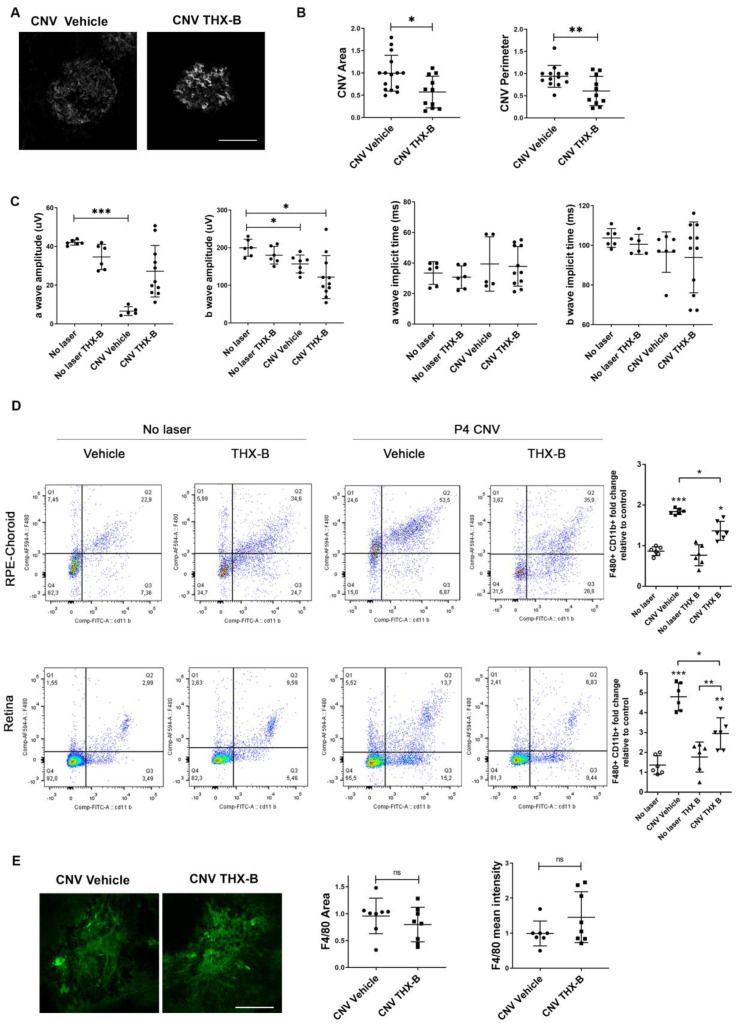
Reduced neovascular phenotype and improved retinal function in p75^NTR^ antagonist-treated wild type mice after laser injury. p75^NTR^ receptor antagonist reduced the area and perimeter of choroidal neovessels and improved retinal functionality in WT CNV mice. (**A**) Representative confocal images of RPE-Choroid flat-mounts of WT mice 7 days after laser, showing immunofluorescence staining with Isolectin IB-4 (grey). Scale bar: 100 µm. (**B**) The area and perimeter of neovessels were quantified with ImageJ FIJI software and represented in the bar graph expressed as units relative to WT CNV vehicle control. Bars denote the mean ± SD from triplicate experiments, *n* = 5 mice/group. (**C**) Amplitudes and implicit times of a- and b-waves from scotopic electroretinograms recorded 7 days after the laser in WT and WT CNV mice injected with THX-B or vehicle. The data show averages of responses of both. Graphs denote the mean ± SD from triplicate experiments, *n* = at least 6 mice/group. (**D**) Representative flow cytometry pseudocolor plots from WT CNV and no laser mice injected with THX-B or vehicle, 4 days after laser. Cells in the gate were quantified and the number of cells in the gate/total cells ratio is represented in the bar graph expressed as units relative to WT no laser vehicle control. Graphs denote the mean ± SD from triplicate experiments, *n* = 6 mice/group. The symbol correspond: No laser (white circle), CNV Vehicle (filled square), No laser THX B (filled triangle up), CNV THX B (filled triangle down). The asterisks show statistical differences respect to control. * *p* < 0.05, ** *p* < 0.01, *** *p* < 0.001. (**E**) Representative confocal images of RPE-Choroid flat-mounts of WT mice 4 days after laser, showing immunofluorescence staining with F4/80 (green). Scale bar: 200 µm. The area and the F4/80 fluorescence intensity were quantified with ImageJ FIJI software and represented in the bar graph expressed as units relative to WT CNV+ vehicle control. Bars denote the mean ± SD from triplicate experiments, *n* = 4 mice/group. ns: non-significant.

## Data Availability

Not applicable.

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
