# Peer review of "Etiological Roles of p75^NTR^ in a Mouse Model of Wet Age-Related Macular Degeneration"

_cells, 2023, doi:10.3390/cells12020297_

Round 1

Reviewer 1 Report

The P75NTR (p75 neurotrophin receptor) belongs to the TNF-α receptor superfamily, which binds neurotrophins and regulates the apoptosis process. In the eyes, increased p75NTR expression was found in ischemic retinopathy and diabetic retinopathy. In this manuscript, the authors tried to define the roles of P75NTR in the development of Choroidal neovascularization by using pharmacological or genetic ablation approaches. They found, the expression of P75NTR significantly upregulated in the early stage of CNV (p4) of the retina and RPE-choroid tissues, when the prior studies found the peak of immune cells invading after laser injury. They also found P75NTR or inhibitor treatment efficiently suppresses the immune cell infiltration (F4/80+ macrophages, as well as GFAP+ expression). Notably, modulation of P75NTR also preserves the vision function implicated by electroretinography (ERG) measurement. Overall, the whole study is well-designed, and the results/conclusion largely make sense. However, the rigor of data and writing require modification in some cases: 

1. Figure 2A, the Y axis of the bar graph should be "Tubulin" not Actin

2. Line 139, change Waco to wako

3. Please provide the size of the sample (N number) for each bar graph.

4. Exchange the figure legends for Figure 3  and Figure 4

5. FACS data in Figur 4a need carefully check. For example, in RPE-choroid, the cell number of Q2 in no laser of P75NTR-KO is significantly more than P4CNV, but shown opposite in the bar graph

Author Response

Reviewer #1
The p75NTR (p75 neurotrophin receptor) belongs to the TNF-α receptor superfamily, which binds neurotrophins and regulates the apoptosis process. In the eyes, increased p75NTR expression was found in ischemic retinopathy and diabetic retinopathy. In this manuscript, the authors tried to define the roles of p75NTR in the development of Choroidal neovascularization by using pharmacological or genetic ablation approaches. They found, the expression of p75NTR significantly upregulated in the early stage of CNV (p4) of the retina and RPE-choroid tissues, when the prior studies found the peak of immune cells invading after laser injury. They also found P75NTR or inhibitor treatment efficiently suppresses the immune cell infiltration (F4/80+ macrophages, as well as GFAP+ expression). Notably, modulation of p75NTR also preserves the vision function implicated by electroretinography (ERG) measurement. Overall, the whole study is well-designed, and the results/conclusion largely make sense. However, the rigor of data and writing require modification in some cases:

Responses to Reviewer #1 Comments
Point 1: Figure 2A, the Y axis of the bar graph should be "Tubulin" not Actin
Response 1: We apologize for this mistake on the graph. The band correspond to β-Actin and the Western blot label has been corrected accordingly on Figure 2A.
Point 2: Line 139, change Waco to wako
Response 2: We have corrected the name of the antibody brand in the text.
Point 3: Please provide the size of the sample (N number) for each bar graph.
Response 3: We thank the reviewer for raising this valuable point. We have added the size number of each experimental group in the figure legends of the revised manuscript.
Point 4: Exchange the figure legends for Figure 3 and Figure 4
Response 4: We thank the reviewer for this observation that has been corrected in the revised manuscript.
Point 5: FACS data in Figur 4A need carefully check. For example, in RPE-choroid, the cell number of Q2 in no laser of P75NTR-KO is significantly more than P4CNV, but shown opposite in the bar graph.
Response 5: We thank the reviewer for this valuable suggestion. We replace the previous representative image of the experimental group ´no laser - P75NTR-KO´ for other more representative Q2 image.

Reviewer 2 Report

The manuscript ‘Etiological roles of p75NTR in a mouse model of wet age-related macular degeneration’ is reports the roles of p75NTR in the recruitment of MPCs, in glial activation and retina function using the laser-induced choroidal neovascularization model. By genetic and pharmacological tools, authors demonstrated that inhibition of p75NTR might benefit the choroidal neovascularization in age-related macular degeneration. Overall, it’s a well-designed and written manuscript. I believe the manuscript is suitable for publication in Cells, after the authors have addressed the following minor comments and questions:

1.       Figure 1B, since no statistically significant was found for P7 CNV to no laser group. ‘n.s’ or p value should be added to the graph. Similar changes should be done for other figures too. Figure 1C-E shows the p75NTR labeling in in F4/80 positive cells, Iba-1 and CX3CR1 positive microglial cells, it will be preferred that authors can label the p75NTR positive cells using arrowhead or star to highlighted them in the representative images.

2.       Figure 2, authors declared that laser induced increased p75NTR expression in P7 CNV compared to no laser control. The representative band in Figure 2A should be replaced since the difference is not obvious. Authors could also show the high magnification images for dual labeling of p75NTR and GFAP to demonstrate the immunological patterns of p75NTR in different cells subtypes.

3.       Figure 3 and Figure 4 were incorrectly placed/mixed up.

4.       In result 3.3, authors claimed that ‘p75NTR KO CNV mice exhibited a significant decrease in MPCs population as compared to WT CNV mice’. The difference should be shown in the graph for p75KO CNV vs WT CNV but not only comparing laser and no laser group.

5.       For Figure 4A and 4B, authors should label the 1day and 7day post laser for clear present.

Author Response

Reviewer #2 ,
The manuscript ‘Etiological roles of p75NTR in a mouse model of wet age-related macular degeneration’ is reports the roles of p75NTR in the recruitment of MPCs, in glial activation and retina function using the laser-induced choroidal neovascularization model. By genetic and pharmacological tools, authors demonstrated that inhibition of p75NTR might benefit the choroidal neovascularization in age-related macular degeneration. Overall, it’s a well-designed and written manuscript. I believe the manuscript is suitable for publication in Cells, after the authors have addressed the following minor comments and questions:

Response to Reviewer 2

Comments
We thank the reviewer for high lining this detail. Here we provide a point-by-point response to your comments

Point 1: Figure 1B, since no statistically significant was found for P7 CNV to no laser group. ‘n.s’ or p value should be added to the graph. Similar changes should be done for other figures too. Figure 1C-E shows the p75NTR labeling in in F4/80 positive cells, Iba-1 and CX3CR1 positive microglial cells, it will be preferred that authors can label the p75NTR positive cells using arrowhead or star to highlighted them in the representative images.
Response 1: We thank the reviewer for raising this important point. The new version of the manuscript includes p values in the graphs where no significant “ns” differences were detected. This p values are embedded in the result section. In addition, arrowheads were added to better visualize co-localization of p75NTR and other markers.

Point 2: Figure 2, authors declared that laser induced increased p75NTR expression in P7 CNV compared to no laser control. The representative band in Figure 2A should be replaced since the difference is not obvious. Authors could also show the high magnification images for dual labeling of p75NTR and GFAP to demonstrate the immunological patterns of p75NTR in different cells subtypes.
Response 2: We thank the reviewer for this important recommendation. We have replaced the representative Western blot image in Figure 2A. Moreover, high magnification images with their respective scale bar were included in the revised version of the manuscript.

Point 3: Figure 3 and Figure 4 were incorrectly placed/mixed up.
Response 3: We apologize for our mistake; in the revised version of the manuscript, we have solved this issue.

Point 4: In result 3.3, authors claimed that ‘p75NTR KO CNV mice exhibited a significant decrease in MPCs population as compared to WT CNV mice’. The difference should be shown in the graph for p75KO CNV vs WT CNV but not only comparing laser and no laser group.
Response 4: We thank the reviewer for this valuable suggestion. We have included the statistical comparison requested on Figure 3A.

Point 5: For Figure 4A and 4B, authors should label the 1day and 7day post laser for clear present.

Response 5: We thank the reviewer for the comment. We have added in the Figure 4A and B the time point tested to improve the experimental understanding and the flow of reading.

Round 2

Reviewer 1 Report

Thank authors for addressing all my points. One more question remains to be addressed: 

1. Figure 3B and Figure 5A use the same image as the WT CNV control. As there are more than images in the quantification bar graph, Please use different WT CNV control for these two figures.

Author Response

Reviewer #1,

Thank authors for addressing all my points. One more question remains to be addressed:

Response to Reviewer 1 Comments

We thank the reviewer for high lining this detail. Here we provide a response to your comment

Point 1: Figure 3B and Figure 5A use the same image as the WT CNV control. As there are more than images in the quantification bar graph, Please use different WT CNV control for these two figures.

Response 1:. We thank the reviewer for this observation that has been corrected in the revised manuscript.